# Cocaine-induced adaptation of dopamine D2S, but not D2L autoreceptors

**Brooks G Robinson[1†]*, Alec F Condon[1†], Daniela Radl[2], Emiliana Borrelli[2], John T Williams[1], Kim A Neve[3,4]**

[1]The Vollum Institute, Oregon Health & Science University, Portland, United States; [2]Department of Microbiology and Molecular Genetics, University of California, Irvine, Irvine, United States; [3]Research Service, VA Portland Health Care System, Portland, United States; [4]Department of Behavioral Neuroscience, Oregon Health & Science University, Portland, United States

**Abstract** The dopamine D2 receptor has two splice variants, D2S (Short) and D2L (Long). In dopamine neurons, both variants can act as autoreceptors to regulate neuronal excitability and dopamine release, but the roles of each variant are incompletely characterized. In a previous study we used viral receptor expression in D2 receptor knockout mice to show distinct effects of calcium signaling on D2S and D2L autoreceptor function (Gantz et al., 2015). However, the cocaine-induced plasticity of D2 receptor desensitization observed in wild type mice was not recapitulated with this method of receptor expression. Here we use mice with genetic knockouts of either the D2S or D2L variant to investigate cocaine-induced plasticity in D2 receptor signaling. Following a single in vivo cocaine exposure, the desensitization of D2 receptors from neurons expressing only the D2S variant was reduced. This did not occur in D2L-expressing neurons, indicating differential drug-induced plasticity between the variants.
DOI: https://doi.org/10.7554/eLife.31924.001

**\*For correspondence:**
robinbro@ohsu.edu

[†]These authors contributed equally to this work

**Competing interests:** The authors declare that no competing interests exist.

## Introduction

Dopamine D2 autoreceptors decrease neuronal excitability when activated by dendrodendritic release of dopamine in the midbrain. The D2 receptor has short (D2S) and long (D2L) splice variants that differ by a 29 amino acid section in the third intracellular loop. While both variants are expressed in dopamine neurons and function effectively as autoreceptors (*Khan et al., 1998*; *Jang et al., 2011*; *Dragicevic et al., 2014*; *Jomphe et al., 2006*; *Neve et al., 2013*), subtle differences have been reported suggesting the variants are not redundant. Agonist induced desensitization of the variants has been shown to differ in multiple reports. The D2S receptor desensitizes and internalizes to a greater degree than D2L (*Liu et al., 1992*; *Itokawa et al., 1996*; *Ito et al., 1999*; *Morris et al., 2007*; *Thibault et al., 2011*). Desensitization of the D2S receptor-dependent activation of G protein-gated inwardly rectifying potassium (GIRK) conductance was also greater and was dependent on the level of intracellular calcium buffering (*Gantz et al., 2015*). Desensitization of the D2 receptor is decreased following in vivo acute drug exposure to ethanol (*Perra et al., 2011*), cocaine (*Dragicevic et al., 2014*; *Gantz et al., 2015*), or L-DOPA (*Dragicevic et al., 2014*). A reduction in desensitization following drug exposure would conceivably result in a more effective autoreceptor (and therefore reduced neuronal excitability) during the next high dopamine situation, which would have widespread effects on dopamine signaling throughout the brain. However, studies using viral expression of single D2 receptor variants in D2-KO mice were inconclusive in determining whether the plasticity induced by cocaine resulted from adaptation of a single variant, or perhaps an altered ratio of splice variant expression/function (*Gantz et al., 2015*). To resolve this issue, the present study used mouse models that express only the D2S (D2L-KO; *Usiello et al., 2000*) or D2L

variant (D2S-KO; *Radl et al., 2013*). The results show that a single treatment of animals with cocaine reduced acute desensitization of the D2S variant. Treatment with cocaine did not alter the expression of either D2 receptor variant, and the desensitization of the D2L variant was not changed. Thus, it appears that plasticity in D2 receptor signaling induced by a single cocaine treatment results from adaptation of the D2S variant.

## Results and discussion

The calcium sensitivity and desensitization of the short and long variants of the D2 receptor were effectively studied with the use of viral expression of D2S and D2L receptors in dopamine neurons of D2-KO mice (*Gantz et al., 2015*). However, the adaptation in D2 receptor desensitization following in vivo cocaine treatment could not be replicated with mice virally expressing D2 receptors (D2S, D2L, or a combination of both) and therefore there was uncertainty about the source of adaptation. To address the issue, D2L-KO and D2S-KO mouse lines were employed. These mice express a single D2 variant at physiological levels (*Usiello et al., 2000*; *Radl et al., 2013*). Whole-cell recordings from dopamine neurons in brain slices containing the substantia nigra pars compacta (SNc) were used to measure desensitization. The selective D2 receptor agonist quinpirole (10 µM) was applied for 5 min and the decline from the peak outward D2-GIRK current was measured after 90 s. This experiment was done in slices taken from untreated and cocaine treated (20 mg/kg 24 hr prior) male mice between 61 and 106 days old. Each genotype was examined using two internal solutions, a weak calcium buffering EGTA (0.1 mM) and a strong calcium buffering BAPTA (10 mM) solution. The comparison of these two internal conditions allows the parsing of processes dependent on and independent of intracellular calcium signaling. Results from wild type littermates from both genotypes were combined as no differences between the groups were seen. D2-GIRK currents from wild type mice desensitized to a greater extent with the weak calcium buffering EGTA internal solution compared to the strong buffering BAPTA internal. With the use of EGTA (and not BAPTA) internal solution the D2-GIRK current desensitized significantly less in cells from animals that were treated with cocaine (*Figure 1A*). This confirms previous results obtained in wild type mice (*Gantz et al., 2015*). The results from experiments with the D2L-KO mice (expressing D2S only) were similar to experiments in wild type mice. The decline in the quinpirole current with EGTA internal was greater than with BAPTA internal and was significantly reduced following in vivo cocaine treatment (*Figure 1B*). In D2S-KO mice (expressing only D2L), some calcium sensitivity in current decline was still present (EGTA currents declined significantly more than BAPTA currents), but treatment with cocaine had no effect in these animals (*Figure 1C*). Representative traces of currents from naïve or cocaine-treated mice of each of the three genotypes in *Figure 1* further demonstrate that D2S, and not D2L, adapts following cocaine exposure. Desensitization is measured 90 s post peak. This initial phase of desensitization is calcium sensitive and differs between D2 receptor variants ($52.7 \pm 5.1\%$ vs. $36.1 \pm 2.9\%$ decline with EGTA internal $p < 0.01$ with Student's t-test, comparison not depicted). In the wild type and D2L-KO genotypes, the initial decline from peak becomes substantially shallower following cocaine exposure indicating an adaptation in a calcium-sensitive aspect of desensitization.

The calcium sensitivity of D2 autoreceptors remains incompletely understood. The L-type calcium channels (*Dragicevic et al., 2014*; *Gantz et al., 2015*) and intracellular calcium stores (*Perra et al., 2011*; *Gantz et al., 2015*) are two sources of calcium that impact D2 receptor signaling. However, there are likely calcium binding proteins that act as intermediaries between the ion and the D2 receptor. Indeed, the intracellular domains of the D2 receptor can bind calmodulin (*Bofill-Cardona et al., 2000*; *Liu et al., 2007*), NCS-1 (*Kabbani et al., 2002*; *Dragicevic et al., 2014*; *Pandalaneni et al., 2015*), and S100B (*Liu et al., 2008*; *Dempsey and Shaw, 2011*) among other potential calcium sensitive proteins. The present results show that while the D2S is more calcium sensitive than D2L, the long variant maintains some calcium sensitivity as evidenced by decreased desensitization when calcium is strongly buffered by the BAPTA internal solution. One explanation is that the calcium sensitivity of the D2 receptor involves multiple sources. Recent imaging of endogenous D2 receptors revealed a clustered and static localization (*Robinson et al., 2017*). It is possible that an array of proteins, potentially including several that are calcium sensitive, forms a complex with D2 receptors modulating their placement and signaling.

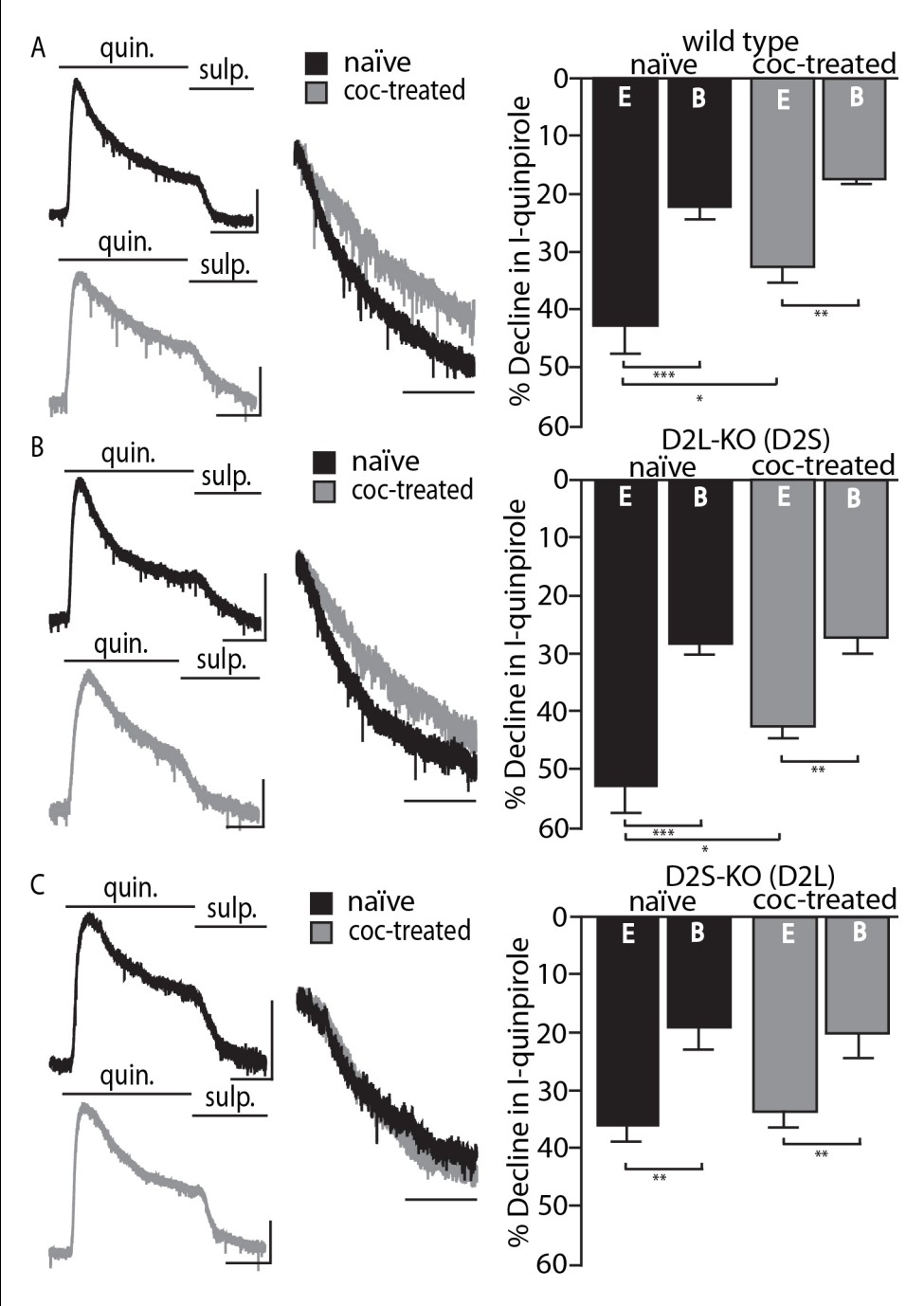

**Figure 1.** Cocaine-induced adaptation in D2 receptor desensitization. Shown on the left are representative traces from naïve and cocaine-treated groups in all genotypes of quinpirole (quin., 10 μM)-induced D2 receptor-GIRK currents (I-quinpirole) reversed by sulpiride (sulp., 600 nM) recorded from dopamine neurons in the SNc using EGTA internal solution. For further comparison, scaled and peak aligned current declines are also shown (horizontal scale bars = 90 s, vertical scale bars = 100 pA). (**A**) In wild type animals there were significant overall effects of internal solution (F(1, 30)=23.57, p<0.001) and cocaine treatment (20 mg/kg, F(1, 30)=4.259, p=0.048) on the decline of I-quinpirole. Post-hoc analyses indicate that in both treatment conditions the decline using BAPTA (**B**) internal was less than that using EGTA (E, p<0.001 for naïve and coc-treated). Additionally, with EGTA internal there was significantly reduced decline following cocaine treatment (p=0.049, n = 7–9 neurons from 5 to 8 mice). (**B**) In mice in which the long isoform of the D2 receptor has been knocked out (D2L-KO), there was an overall effect of internal solution (F(1, 32)=37.09, p<0.001), but not of cocaine treatment (F (1, 32)=2.917, p=0.097). In post

*Figure 1 continued on next page*

*Figure 1 continued*

hoc analyses, there was significantly more decline when using EGTA internal than BAPTA internal in both treatment conditions (p<0.001 for naïve, p=0.002 for cocaine-treated), and the decline was significantly reduced following drug exposure when EGTA internal was used (p=0.021, n = 8–12 neurons from 4 to 7 mice). (**C**) In animals with the short isoform of the D2 receptor knocked out (D2S-KO), there was an overall effect of internal solution (F(1, 28)=21.24, p<0.001) with EGTA currents declining significantly more that those of BAPTA in both treatment conditions (p=0.001 for naïve, p=0.007 for coc-treated, n = 7–9 neurons from 4 to 7 mice). Cocaine treatment caused no change in desensitization of the D2-GIRK current in this genotype. Comparisons were done using two-way ANOVAs followed by Fisher's LSD when p<0.05. *p<0.05, **p<0.01, ***p<0.001.
DOI: https://doi.org/10.7554/eLife.31924.002

The following source data is available for figure 1:

**Source data 1.**
DOI: https://doi.org/10.7554/eLife.31924.003

---

Adaptation in one D2 receptor variant but not the other is relevant to studies in humans that have identified two intronic single nucleotide polymorphisms (SNPs) in the DRD2 gene that cause an increase in D2L expression relative to D2S. These SNPs have been found to be associated with cocaine (*Moyer et al., 2011*; *Levran et al., 2015*), alcohol (*Sasabe et al., 2007*), and opioid abuse (*Clarke et al., 2014*; *Levran et al., 2015*). This has been interpreted mainly as altering the balance between striatal postsynaptic D2 receptor activation (assumed to be D2L) vs. dopamine neuron autoreceptor activation (assumed to be D2S), but our data call into question this hypothesis about the dichotomous function of the D2 receptor splice variants (current results; *Neve et al., 2013*; *Gantz et al., 2015*). However, because dopamine neurons can express either variant, the ratio of D2 autoreceptor variants could very well be important in drug related behaviors. Increased presence of the D2L variant as the autoreceptor would alter the dopamine reward circuitry because this variant desensitizes less (suggesting it is a more efficacious autoreceptor) and does not appear to adapt following a single cocaine exposure. Furthermore, following cocaine exposure desensitization of the D2S variant adapts to resemble the D2L variant. In this assay, preexisting prevalence of the D2L variant would resemble a previous cocaine experience.

The present results are in some respects at odds with the previous study that used viral D2 receptor variant expression and found no calcium-dependent desensitization of D2L and no cocaine-induced adaptation of either variant (*Gantz et al., 2015*), rather than decreased calcium-dependent desensitization of D2L and selective cocaine-induced adaptation of D2S. A possible explanation is that virally transduced D2 receptors have altered expression levels (often superphysiological) resulting in reduced sensitivity to adaptive mechanisms. It must be noted, however, that we cannot exclude the possibility of developmental abnormalities in mice that are constitutive knockouts of D2S, D2L or both D2 variants. One clear advantage of the D2S- and D2L-KO mice used in this study over viral receptor expression is the maintenance of physiological levels of expression. *Figure 2* shows the current densities in pA/pF (a reliable measure of expression) of D2-GIRK currents from wild type, D2L-KO, and D2S-KO animals. While viral receptor expression is not well controlled and often much higher than physiological levels (*Gantz et al., 2015*), the KO animals were in the same range, or lower (*Figure 2B–C*) compared to wild type (*Figure 2A*). The previously shown trend of GIRK currents having increased amplitude upon strong calcium buffering is clearly present (*Figure 2A–B*), however in the D2S-KO genotype, there is no significant difference between current densities in the EGTA and BAPTA groups (*Figure 2C*). Additionally, a comparison between D2L-KO and D2S-KO groups reveals an overall significant effect of genotype on current density (F(1, 28) =5.27, p=0.029, two-way ANOVA). This effect is likely not due to differences in GIRK channel expression because GABA$_B$-GIRK current density was not different between the groups (in pA/pF D2L-KO EGTA 16.3 ± 2.3, BAPTA 18.2 ± 3.6 vs. D2S-KO EGTA 16.0 ± 0.6, BAPTA 18.8 ± 1.2, p>0.05 two-way ANOVA, data not shown). Because of the similarity of D2S-mediated responses to responses of neurons from wild type mice, despite the presence of mRNA for both subtypes (*Khan et al., 1998*; *Jang et al., 2011*; *Dragicevic et al., 2014*) and the ability of both to activate GIRKs (*Jomphe et al., 2006*; *Neve et al., 2013*; *Gantz et al., 2015*), it was speculated that D2L function is in some way occluded under basal conditions (*Gantz et al., 2015*). The lower current density for D2L in this study, despite similar levels of expression of the splice variants in these two lines

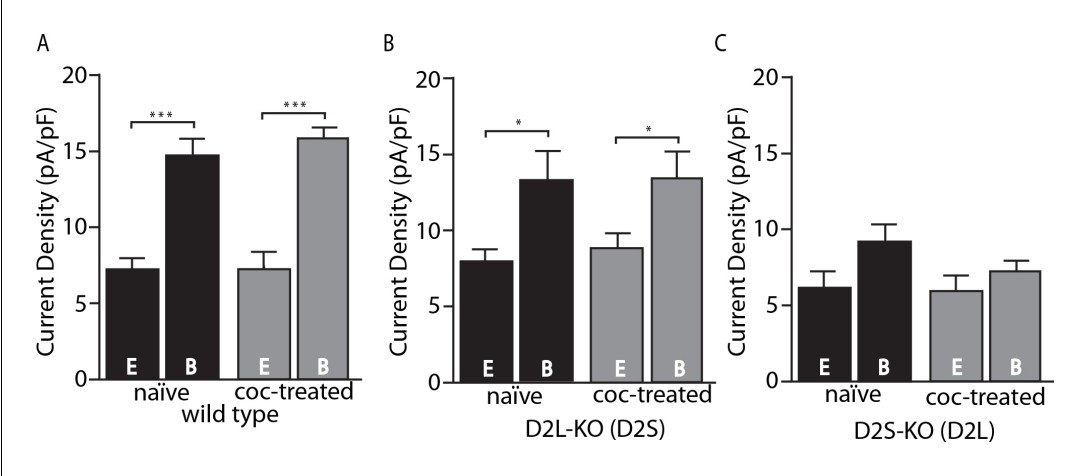

**Figure 2.** Amplitude of D2-GIRK currents. (**A**) In wild type mice, there was an overall significant effect of internal solution on D2-GIRK current density (pA/pF, $F_{(1, 30)}$=76.32, $p<0.001$). The current density was greater when using the strong calcium buffering BAPTA (B) internal solution compared with the weak buffering EGTA (E) internal solution ($p<0.001$ for naïve and coc-treated, n = 7–11 neurons from 5 to 7 mice). There was no difference between naïve and cocaine-treated groups. (**B**) From D2L-KO mice, there was an overall significant effect of internal solution ($F_{(1, 32)}$=12.85, $p=0.001$). The current density of the BAPTA group was significantly larger than that of EGTA in both naïve and coc-treated conditions ($p=0.016$ for naïve, $p=0.016$ for coc-treated, n = 7–12 neurons from 4 to 7 mice), however there was no difference between treatment conditions. (**C**) In D2S-KO mice, there was no significant effect of internal solution or drug treatment (n = 7–9 neurons from 5 to 7 mice). Analyses were done by two-way ANOVAs followed by Fisher's LSD when $p<0.05$. $*p<0.05$, $**p<0.01$, $***p<0.001$.
DOI: https://doi.org/10.7554/eLife.31924.004
The following source data is available for figure 2:

**Source data 1.**
DOI: https://doi.org/10.7554/eLife.31924.005

of mice (D. Radl, M. Chiacchiaretta, R. Lewis, K. Brami-Cherrier, L. Arcuri, & E. Borrelli, manuscript in preparation), is consistent with this notion. Alternatively, the lower current density for D2L and lack of calcium-dependent desensitization could both be explained if the D2L receptor were constitutively desensitized.

Importantly, the desensitization of GABA$_B$ receptor-induced GIRK currents with baclofen (30 µM) was not sensitive to the two different internal solutions and did not change in any genotype following cocaine exposure (*Figure 3A–C*). Thus, the calcium sensitive desensitization and the cocaine-induced adaptations are specific to the D2 receptor and not general to Gi-coupled GPCRs or GIRK channels.

## Concluding remarks

A genetic strategy was used to address the outstanding question of how the desensitization of dopamine D2 autoreceptors adapts following in vivo cocaine exposure. Mice with either the long or short splice variant of the D2 receptor constitutively knocked out were employed to show that the cocaine-induced decrease in D2 receptor desensitization occurs when D2S is the only variant expressed. This mimics the wild type phenotype and does not occur when only the D2L variant is expressed.

## Materials and methods

### Animals/Key Resources

All experiments were done in strict accordance with the Institutional Animal Care and Use Committees (IACUCs) at the VA Portland Health Care System (VAPORHCS) and Oregon Health and Science University (OHSU). Mice with genetic deletion of the long splice variant of the dopamine D2 receptor (D2L-KO or D2L-/-) and the short variant of the D2 receptor (D2S-KO or D2S-/-) were used in this study in addition to wild types from each of these lines. Mice originated from the Borrelli laboratory

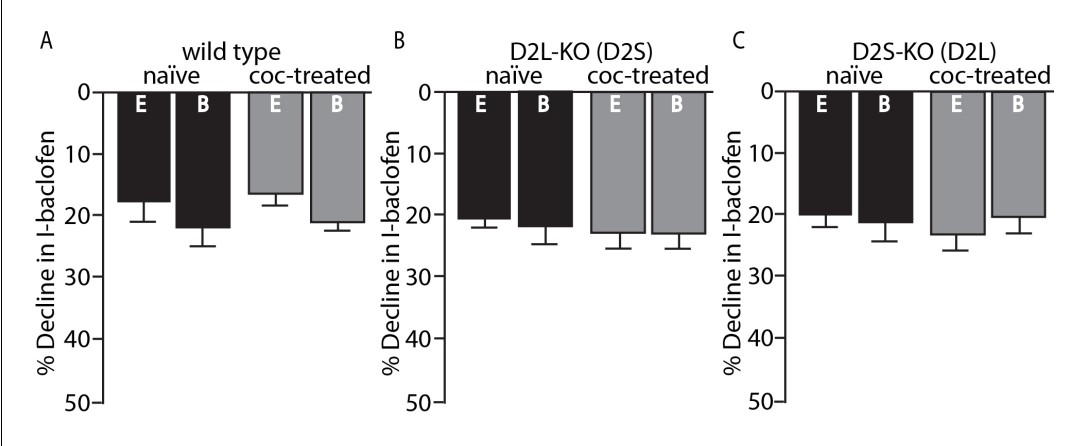

**Figure 3.** No change in GABA$_B$ desensitization following cocaine exposure. GABA$_B$ receptor decline/desensitization was measured by bath application of baclofen (30 µM) for ~5 min. There were no significant effects of EGTA (E, 0.01 mM) versus BAPTA (B, 10 mM) internal solution or treatment condition (naïve versus coc-treated) on the decline in GABA$_B$-GIRK currents (I-baclofen) from (A) wild type mice, (B) D2L-KO mice, or (C) D2S-KO mice (p<0.05, n = 5–9 neurons from 3 to 6 mice). Analyses were done using two-way ANOVAs.

DOI: https://doi.org/10.7554/eLife.31924.006

The following source data is available for figure 3:

**Source data 1.**

DOI: https://doi.org/10.7554/eLife.31924.007

(UC Irvine). For further information on creation of lines refer to *Usiello et al. (2000)* and *Radl et al. (2013)*.

### Slice preparation and electrophysiology

For detailed electrophysiological methods, please refer to *Gantz et al. (2015)*.

## Acknowledgements

This work was supported by NIH funding DA04523 (JTW), F32 DA038456 (BGR), and DA033554 (EB), INSERM (EB), and by Merit Review Award BX003279 from the US Department of Veterans Affairs, Veterans Health Administration, Office of Research and Development, Biomedical Laboratory Research, and Development (KAN).

## Additional information

### Funding

| Funder | Grant reference number | Author |
| --- | --- | --- |
| National Institutes of Health | F32 DA038456 | Brooks G Robinson |
| National Institutes of Health | DA033554 | Emiliana Borrelli |
| Institut National de la Santé et de la Recherche Médicale | | Emiliana Borrelli |
| National Institutes of Health | DA04523 | John T Williams |
| National Institute on Drug Abuse | T32DA007262 | John T Williams |
| U.S. Department of Veterans Affairs | BX003279 | Kim A Neve |

The funders had no role in study design, data collection and interpretation, or the decision to submit the work for publication.

## Author contributions
Brooks G Robinson, Conceptualization, Data curation, Formal analysis, Funding acquisition, Validation, Investigation, Methodology, Writing—original draft, Project administration, Writing—review and editing; Alec F Condon, Conceptualization, Data curation, Formal analysis, Validation, Investigation, Methodology, Writing—review and editing; Daniela Radl, Resources, Methodology, Writing—review and editing; Emiliana Borrelli, Conceptualization, Resources, Funding acquisition, Methodology, Writing—review and editing; John T Williams, Conceptualization, Resources, Funding acquisition, Methodology, Project administration, Writing—review and editing; Kim A Neve, Conceptualization, Resources, Supervision, Funding acquisition, Methodology, Project administration, Writing—review and editing

## Author ORCIDs
Brooks G Robinson http://orcid.org/0000-0001-5020-531X
Alec F Condon http://orcid.org/0000-0003-2655-2121
John T Williams http://orcid.org/0000-0002-0647-6144
Kim A Neve http://orcid.org/0000-0003-0109-7345

## Ethics
Animal experimentation: This study was performed in strict accordance with the Institutional Animal Care and Use Committees at the VA Portland Health Care System (VAPORHCS) protocol 3681-16 and Oregon Health & Science University (OHSU) protocol IP00000160.

## Decision letter and Author response
Decision letter https://doi.org/10.7554/eLife.31924.010
Author response https://doi.org/10.7554/eLife.31924.011

# Additional files

## Supplementary files
• Transparent reporting form
DOI: https://doi.org/10.7554/eLife.31924.008

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
