## [Decision Letter]

Thank you for submitting your article "Cocaine-induced adaptation of dopamine D2S, but not D2L autoreceptors" for consideration by *eLife*. Your article has been reviewed by three peer reviewers, and the evaluation has been overseen by Sacha Nelson as a Reviewing Editor and Eve Marder as the Senior Editor. The following individuals involved in review of your submission have agreed to reveal their identity: David M Lovinger (Reviewer #1); Louis-Eric Trudeau (Reviewer #2).

The reviewers have discussed the reviews with one another and the Reviewing Editor has drafted this decision to help you prepare a revised submission.

Summary:

The present manuscript by Robinson and colleagues is a useful and informative extension of the author's previous paper published in *eLife* in 2015. The work takes advantage of mice in which the short or long isoform of the D2 receptor are separately expressed due to selective knockout of one of the two. The authors' main conclusion is that the calcium-dependent desensitization of the D2R and the reduction of such desensitization after cocaine treatment is similar in dopamine neurons expressing only the D2S isoform and in WT mice expressing both isoforms. However, both of these processes are attenuated or absent in dopamine neurons expressing only the D2L. The work is solid and the conclusions sound.

Essential revisions:

The reviewers have requested some additional statistical analyses and some textual changes for clarity. The original reviews are included below.

*Reviewer #1:*

The investigators report differences in cocaine-induced alterations in desensitization of the short and long variants of the D2 receptor in neurons of the ventral tegmental area. Mice that lack one or the other receptor variant are used in pharmacological and electrophysiological experiments. This work follows up on a previous study from this group showing differences in calcium-dependent desensitization and in the cocaine effect between the different receptor sub-forms using a viral overexpression technique. The effect of a single in vivo dose of cocaine is clearly lost in mice that express only D2L receptors, and the desensitization is larger under conditions of lower calcium buffering independent of cocaine treatment. Interestingly, cocaine and calcium buffering do not alter desensitization of GABAB receptor effects mediated through the same GIRK channel effector. The data also appear to show that D2-activated GIRK current density is lower in D2L-expressing neurons, and that calcium buffering and cocaine have no influence on this measure in these mice, while stronger buffering increases current density in VTA neurons from mice expressing D2S or a mixed receptor population. The authors also nicely extend their previous findings by using the sub-form knockout mice instead of viral expression which keeps the current densities within a relatively normal range. The scope of the study is very limited, but in general the findings are clear.

There are only a couple of additional analyses that would further clarify the findings:

1) In Figure 1, the authors should determine if the agonist-induced decline in current amplitude differs after cocaine treatment in the BAPTA recording condition. This appears to be the case in the wild type mice (although it did not differ in WT mice in the original paper) but not in the two knockouts, but statistical analysis of these data is needed.

2) The findings in Figure 2 appear to indicate lower current density when D2L is the only sub-form expressed, especially in the BAPTA condition. A statistical comparison of these current densities across genotypes should be performed. This difference was not observed in the previous study when viral overexpression was used. Is this due to a change in receptor or GIRK density? Does it suggest that D2L receptors are already "pre-desensitized" and thus insensitive to the calcium effect? Also, since the density and calcium-dependent desensitization are both altered with this receptor, it might be good to correlate current density with desensitization across all cells in all genotypes in the EGTA and BAPTA conditions to determine to what extent these variables are related.

*Reviewer #2:*

The present manuscript by Robinson and colleagues is a useful and informative extension of the author's previous paper published in *eLife* in 2015. The work takes advantage of mice in which the short or long isoform of the D2 receptor are separately expressed due to selective knockout of one of the two. The authors' main conclusion is that the calcium-dependent desensitization of the D2R and the reduction of such desensitization after cocaine treatment is similar in dopamine neurons expressing only the D2S isoform and in WT mice expressing both isoforms. However, both of these processes are attenuated or absent in dopamine neurons expressing only the D2L. The work is solid and the conclusions sound.

The Introduction needs one or two sentences to provide a general context for the work, for example explaining why changes in D2R desensitization after drug treatment is something that is of particular interest.

At the beginning of the Results and Discussion section, when the authors refer to the use of BAPTA and EGTA internal solutions, they should immediately explain the purpose of comparing these two recording conditions.

The authors seem to have analyzed all of their data using a one-way ANOVA. Since two factors are included in each experiment (type of buffer and presence or absence of cocaine treatment), a two-way ANOVA is needed.

The authors state in the result that the calcium-dependent desensitization of D2S is larger than that of D2L. An appropriate statistical comparison is needed to make this statement.

The authors need to discuss the results of the present work in relationship to their previous work using viral expression of D2S or D2L, and also in relationship to previous work by other groups. Importantly, they need to comment on the similarities and differences in the conclusions of their two studies. For example, in their previous study, cocaine treatment did not reduce calcium-dependent D2R desensitization in neurons expressing only the D2S: this is seemingly contradictory to the present results and thus needs to be discussed.

*Reviewer #3:*

Robinson et al. address the contribution of the different D2R splice variants to the adaptation of dopamine neuron autoreceptor function following cocaine exposure. Counter to the previous report (Gantz et al. 2015), they find that D2S, not D2L, undergoes cocaine-dependent alteration. The results are clearer because the D2R's are separated by selective knockout, rather than global D2 knock out and viral restoration, so developmental confounds are reduced, and expression remains physiological, as demonstrated in Figure 2 (showing that there is no impact of cocaine treatment on quinpirole-induced K^+^ currents D2-GIRK currents). Furthermore, the impact was specific to D2 receptors as there was no impact of genotype or treatment on GABA-B receptor mediated desensitization. It should both be acknowledged 1) that the data in Figure 2 and Figure 3 show lack of developmental perturbations, and yet 2) the mice are constitutive knockouts (of either D2S or D2L) so that developmental contributions cannot be excluded. Notwithstanding this comment, the present results are very clear, and constitute a significant advance in identifying differential autoreceptor roles of the D2R splice variants.

Abstract: Does not state directly the precedent for the Research Advance. The relevant statement regarding the previous study (Gantz et al., 2015) was, "Previous in vivo cocaine exposure removed calcium-dependent D2 autoreceptor desensitization in wild type, but not D2S-only mice… implying a functional role for the co-expression of D2S and D2L auto receptors."

Introduction: The statement that needs to be referenced in the Abstract regarding the Research Advance is made in the Introduction, namely, "However, studies using viral expression of single D2 receptor variants in D2-KO mice were inconclusive in determining whether the plasticity induced by cocaine resulted from adaptation of a single variant, or perhaps an altered ratio of splice variant expression/function (Gantz et al., 2015)."

Figure 1: The offset of the traces makes interpretation more difficult. Gantz et al. (2015) presented traces superimposed which are clearer, and given that this Research Advance follows the same methodology, the display of the data should be the same. The offset also obscures differences in the timing of responses, which are puzzling; for instance, the cocaine-treated slices show onset of supliride action earlier than naive in Panel B, but the opposite (later) in Panel C.

---

## [Author Response]

Reviewer #1:[…] There are only a couple of additional analyses that would further clarify the findings:1) In Figure 1, the authors should determine if the agonist-induced decline in current amplitude differs after cocaine treatment in the BAPTA recording condition. This appears to be the case in the wild type mice (although it did not differ in WT mice in the original paper) but not in the two knockouts, but statistical analysis of these data is needed.

This comparison was done in the original analysis, but was not specifically mentioned, as there was no significant difference between the BAPTA naïve and BAPTA cocaine groups. This result is now included in the first paragraph of the Results and Discussion section.

2) The findings in Figure 2 appear to indicate lower current density when D2L is the only sub-form expressed, especially in the BAPTA condition. A statistical comparison of these current densities across genotypes should be performed. This difference was not observed in the previous study when viral overexpression was used. Is this due to a change in receptor or GIRK density? Does it suggest that D2L receptors are already "pre-desensitized" and thus insensitive to the calcium effect? Also, since the density and calcium-dependent desensitization are both altered with this receptor, it might be good to correlate current density with desensitization across all cells in all genotypes in the EGTA and BAPTA conditions to determine to what extent these variables are related.

This analysis and discussion has been added in the fourth paragraph of the Results and Discussion section. There was a significant difference found between the D2L-KO and D2S-KO groups in current density. This is likely not due to GIRK channel expression/density as the current densities of GABAB-GIRK currents were identical.

Reviewer #2:[…] The Introduction needs one or two sentences to provide a general context for the work, for example explaining why changes in D2R desensitization after drug treatment is something that is of particular interest.

A sentence regarding general context has been added in the Introduction.

At the beginning of the Results and Discussion section, when the authors refer to the use of BAPTA and EGTA internal solutions, they should immediately explain the purpose of comparing these two recording conditions.

The reason for comparing EGTA and BAPTA internal solutions has been added to the first paragraph of the Results and Discussion section.

The authors seem to have analyzed all of their data using a one-way ANOVA. Since two factors are included in each experiment (type of buffer and presence or absence of cocaine treatment), a two-way ANOVA is needed.

The analyses have been redone using two-way ANOVAs.

The authors state in the result that the calcium-dependent desensitization of D2S is larger than that of D2L. An appropriate statistical comparison is needed to make this statement.

This statistical comparison has been added to the first paragraph of the Results and Discussion section.

The authors need to discuss the results of the present work in relationship to their previous work using viral expression of D2S or D2L, and also in relationship to previous work by other groups. Importantly, they need to comment on the similarities and differences in the conclusions of their two studies. For example, in their previous study, cocaine treatment did not reduce calcium-dependent D2R desensitization in neurons expressing only the D2S: this is seemingly contradictory to the present results and thus needs to be discussed.

A short discussion has been added on the discrepancy between the previous and current studies in the fourth paragraph of the Results and Discussion section. Our hypothesis is that viral overexpression of D2S may produce results not seen when the receptors are constitutively expressed at levels similar to native D2 receptors.

Reviewer #3:Robinson et al. address the contribution of the different D2R splice variants to the adaptation of dopamine neuron autoreceptor function following cocaine exposure. Counter to the previous report (Gantz et al. 2015), they find that D2S, not D2L, undergoes cocaine-dependent alteration. The results are clearer because the D2R's are separated by selective knockout, rather than global D2 knock out and viral restoration, so developmental confounds are reduced, and expression remains physiological, as demonstrated in Figure 2 (showing that there is no impact of cocaine treatment on quinpirole-induced K^+^ currents D2-GIRK currents). Furthermore, the impact was specific to D2 receptors as there was no impact of genotype or treatment on GABA-B receptor mediated desensitization. It should both be acknowledged 1) that the data in Figure 2 and Figure 3 show lack of developmental perturbations, and yet 2) the mice are constitutive knockouts (of either D2S or D2L) so that developmental contributions cannot be excluded. Notwithstanding this comment, the present results are very clear, and constitute a significant advance in identifying differential autoreceptor roles of the D2R splice variants.

Discussion of viral reconstitution versus genetic KO, including the possibility of developmental compensation, has been added throughout the fourth paragraph of the Results and Discussion section.

Abstract: Does not state directly the precedent for the Research Advance. The relevant statement regarding the previous study (Gantz et al., 2015) was, "Previous in vivo cocaine exposure removed calcium-dependent D2 autoreceptor desensitization in wild type, but not D2S-only mice… implying a functional role for the co-expression of D2S and D2L auto receptors."Introduction: The statement that needs to be referenced in the Abstract regarding the Research Advance is made in the Introduction, namely, "However, studies using viral expression of single D2 receptor variants in D2-KO mice were inconclusive in determining whether the plasticity induced by cocaine resulted from adaptation of a single variant, or perhaps an altered ratio of splice variant expression/function (Gantz et al., 2015)."

A sentence has been added to the Abstract regarding the precedent for this research advance.

Figure 1: The offset of the traces makes interpretation more difficult. Gantz et al. (2015) presented traces superimposed which are clearer, and given that this Research Advance follows the same methodology, the display of the data should be the same. The offset also obscures differences in the timing of responses, which are puzzling; for instance, the cocaine-treated slices show onset of supliride action earlier than naive in Panel B, but the opposite (later) in Panel C.

Figure 1 has been changed to eliminate the offset currents. The decline aspects of the currents have been scaled and peak-aligned for a clearer comparison. The kinetics of both agonist and antagonist onset during the bath application of drugs in electrophysiology experiments in slice preparations are variable, making it difficult to simply overlay the full traces and also difficult to interpret apparent kinetic differences among groups. These examples were chosen because they were representative of the data for the level of desensitization during agonist exposure.